# Novel prognostic biomarkers of pouchitis after ileal pouch-anal anastomosis for ulcerative colitis: Neutrophil-to-lymphocyte ratio

**Yu Nishida[1], Shuhei Hosomi[1]\*, Hirokazu Yamagami[2], Koji Fujimoto[1], Rieko Nakata[1], Shigehiro Itani[1], Yuji Nadatani[1], Shusei Fukunaga[1], Koji Otani[1], Fumio Tanaka[1], Yasuaki Nagami[1], Koichi Taira[1], Noriko Kamata[1], Toshio Watanabe[1], Yasuhito Iseki[3], Tatsunari Fukuoka[3], Masatsune Shibutani[3], Hisashi Nagahara[3], Satoko Ohfuji[4], Yasuhiro Fujiwara[1]**

1 Department of Gastroenterology, Osaka City University Graduate School of Medicine, Osaka, Japan, 2 Department of Gastroenterology, Ishikiri Seiki Hospital, Higashi-Osaka, Japan, 3 Department of Gastroenterological surgery, Osaka City University Graduate School of Medicine, Osaka, Japan, 4 Department of Public health, Osaka City University Graduate School of Medicine, Osaka, Japan

\* m1265271@med.osaka-cu.ac.jp

**Data Availability Statement:** All relevant data are within the manuscript and its Supporting Information files.

## Abstract

### Objectives

Pouchitis is a major complication after restorative proctocolectomy with ileal pouch-anal anastomosis (IPAA) in patients with ulcerative colitis (UC). Although there have been many investigations of the neutrophil-to-lymphocyte ratio (NLR) in various diseases, its role in predicting the development of pouchitis remains unclear. We aimed to evaluate the clinical utility of the NLR for predicting the development of pouchitis after IPAA in UC patients.

### Materials and methods

UC patients who underwent IPAA at Osaka City University Hospital between May 2006 and March 2019 were included. The incidence of pouchitis was estimated using the Kaplan-Meier method. Potential preoperative, intraoperative, and postoperative predictors for pouchitis, including various demographic and clinical variables, were analyzed. The combined impact of the NLR and other known prognostic factors were investigated using Cox proportional hazard regression with inverse probability of treatment weighting (IPTW).

### Results

Forty-nine patients with UC who underwent IPAA were included. The median follow-up period was 18.3 months (interquartile range: 10.7–47.2 months). Eighteen patients (36.7%) developed pouchitis. The incidence of pouchitis was 19.2%, 32.6%, and 45.9% at 1, 2, and 5 years, respectively. NLR was significantly associated with the development of pouchitis in the univariate Cox regression analysis (hazard ratio (HR), 1.14; 95% confidence interval (CI), 1.01–1.28; P = 0.03). The NLR cutoff value of 2.15 was predictive of the development of pouchitis according to receiver operating characteristic analysis (specificity: 67.7%,

**Funding:** The authors received no specific funding for this work.

**Competing interests:** The authors have declared that no competing interests exist.

sensitivity: 72.2%). The incidence of pouchitis was significantly lower in the low NLR group than that in the high NLR group (P = 0.01, log-rank test). Cox regression analyses using IPTW also identified NLR as a prognostic factor for the development of pouchitis by statistically adjusting for background factors (HR, 3.60; 95% CI, 1.31–9.89; P = 0.01).

## Conclusions

NLR may be a novel and useful indicator for predicting the development of pouchitis after IPAA in UC and should be introduced in clinical practice.

## Introduction

Restorative proctocolectomy with ileal pouch-anal anastomosis (IPAA) has become a standard surgical procedure for medically intractable ulcerative colitis (UC), colitis-associated dysplasia, and cancer [1]. The most common long-term complication of this surgery is pouchitis, with a cumulative prevalence of 15%–50% [2–4], which is one of the major factors reducing the postoperative quality of life (QOL) in patients with UC [5, 6].

Although the main cause of pouchitis is unclear, it is more prevalent in patients with UC than in those with familial adenomatous polyposis [1, 7]. Despite conflicting results, the development of pouchitis in patients with UC, as reported in several studies, has been linked to various factors, including primary sclerosing cholangitis [2, 4], other extraintestinal manifestations of inflammatory bowel disease [4, 8, 9], young age at UC diagnosis [4], preoperative terminal ileal inflammation [10, 11], extensive colonic disease [10], presence of interleukin-1 receptor antagonist gene allele 2 [12], total steroid dose of > 10000 mg [13], use of infliximab [14], neutrophil percentage of > 65% [13], and presence of perinuclear antineutrophil cytoplasmic antibodies [15, 16]. *Interleukin-1 receptor antagonist gene allele 2* and perinuclear antineutrophil cytoplasmic antibodies have been reported to be useful markers for predicting the development of pouchitis; however, measuring these markers is costly and not covered by medical insurance in Japan. Easily accessible and low-cost markers are required to predict pouchitis in patients with IPAA.

A growing body of evidence has suggested that the neutrophil-to-lymphocyte ratio (NLR) is not only an easily accessible laboratory test, but also a useful predictive parameter for various types of cancer [17–20], rheumatoid arthritis [21], coronavirus disease 2019 [22], and coronary heart disease [23, 24]. Regarding inflammatory bowel disease, several studies have reported an association between the NLR and UC disease activity [25, 26]. We previously reported the utility of NLR as a useful prognostic marker for predicting the long-term outcomes in patients with UC treated with infliximab or tacrolimus therapy [27, 28]. Lorenzo *et al.* reported that the NLR played a promising role as an early predictor of therapeutic response to anti-tumor necrosis factor (TNF) therapy in patients with UC [29]. However, no studies have tested the value of this parameter in predicting the development of pouchitis after IPAA in patients with UC. Therefore, our study aimed to evaluate the clinical utility of the NLR for predicting the development of pouchitis after IPAA in patients with UC.

## Materials and methods

### Patients

All patients with UC who underwent IPAA at Osaka City University Hospital between May 2006 and March 2019 were included. In our facility, patients underwent one-, two- or three-stage surgeries considered by the patient's preoperative conditions, such as massive bleeding,

rectal inflammation, vital signs, anemia, nutritional status, and age. The two-stage operation starts with total proctocolectomy and IPAA, along with a diversion ileostomy construction in the first stage and ends with ileostomy closure in the second stage. The three-stage operation starts with subtotal colectomy, along with ileostomy construction and sigmoid mucous fistula formation, in the first stage, followed by remnant proctocolectomy and IPAA, with a reconstruction of the ileostomy in the second stage. The procedure is completed with ileostomy closure in the third stage [30]. Patients with one-stage surgery were excluded because we analyzed postoperative NLR.

## Evaluation

All patients were followed up with a physical examination and a blood test. The differential white blood cell (WBC) count was analyzed using an XE-5000 hematology analyzer (Sysmex, Kobe, Japan), as per the manufacturer's protocol. In patients undergoing one-stage surgery, the NLR was calculated from a blood sample measured before IPAA by dividing the absolute neutrophil count by the absolute lymphocyte count. Patients were followed up from the time of IPAA to the onset of pouchitis, loss to follow-up, or until the end of March 2020. The NLR was calculated from a blood sample measured before stoma closure. Patients were followed up from the time of stoma closure to the onset of pouchitis, loss to follow-up, or until the end of March 2020.

## Diagnosis of pouchitis

The pouchitis disease activity index (PDAI) is a 19-point index of pouchitis activity based on clinical symptoms as well as endoscopic and histologic findings [31]. In this study, pouchitis was diagnosed based on the modified pouchitis disease activity index (mPDAI) score using a combination of clinical symptoms and endoscopic examination. An mPDAI score of $\geq 5$ points was used to define pouchitis in this study [32].

## Study endpoints

The primary outcome measure of this study was the onset of pouchitis. Potential preoperative, intraoperative, and postoperative predictors for pouchitis, including various demographic and clinical variables, were analyzed.

## Statistical analysis

Continuous variables are presented as medians and interquartile ranges. The differences in clinical characteristics were compared using either the chi-square test or Fisher's exact test for categorical variables and the Mann-Whitney U-test for continuous variables. Receiver operating characteristic (ROC) curves were plotted so that the area under the ROC curve could be calculated. Optimal cutoff values were determined according to the Youden criterion, which marks the point on a ROC curve where "sensitivity + specificity– 1" is maximal [33]. The cumulative incidence of pouchitis was illustrated using a Kaplan-Meier plot. Differences in the survival curves were assessed using the log-rank test. Furthermore, continuous values of laboratory data were evaluated using the Cox proportional hazard model. Data are presented as hazard ratios (HRs) with 95% confidence intervals (CIs). An IPTW analysis was applied to each observation in the Cox model to assess the relationship between the NLR and the development of pouchitis. The IPTW analysis was derived using propensity scores on all observations before matching to reduce selection bias by statistically adjusting for background factors [34]. Variables included in the IPTW analysis were age, sex, disease location, and history of anti-TNF therapy.

A P-value of < 0.05 was considered statistically significant. All statistical analyses were performed with EZR (Saitama Medical Center, Jichi Medical University), a graphical user interface for R (The R Foundation for Statistical Computing, version 2.13.0). More precisely, it is a modified version of R commander (version 1.6–3) that includes statistical functions frequently used in biostatistics.

### Ethical considerations

This study was approved by the Osaka City University Hospital Certified Review Board; (no. 4291), which waived the requirement for written informed consent because the analysis used anonymized clinical data that were retrospectively obtained after each patient agreed to receive the treatment. All data were fully anonymized before we accessed them. Nevertheless, all patients were notified of the content and information of this study and given the opportunity to refuse participation. None of the patients refused participation. This study followed the Ethical Guidelines for Medical and Health Research Involving Human Subjects established by the Ministry of Education, Culture, Sports, Science and Technology and the Ministry of Health, Labor and Welfare in Japan.

## Results

### Study subjects

Overall, we included 79 patients who underwent IPAA for UC during the study period. Twenty-nine patients were excluded due to a lack of data on differential WBC count. One patient with one-stage surgery was excluded. Forty-nine patients were retrospectively reviewed. Of those patients, 30 patients underwent IPAA for disease refractory to medication, 9 patients for dysplasia or cancer, 3 patients for perforation, 3 patients for toxic megacolon, 2 patients for colonic strictures, and 2 patients for massive bleeding.

The median follow-up period was 18.3 months (interquartile range: 10.7–47.2 months). Eighteen patients (36.7%) developed pouchitis. The incidence of pouchitis was 19.2%, 32.6%, and 45.9% at 1, 2, and 5 years, respectively (Fig 1). The demographic characteristics of the patients are summarized in Table 1.

### Risk factors for pouchitis

NLR was significantly associated with the development of pouchitis according to univariate Cox regression analysis (HR, 1.14; 95% CI, 1.01–1.28; P = 0.03). No other clinical variables such as age, sex, disease duration, age at onset, or disease location have shown a statistically significant association with pouchitis development (Table 2).

When the NLR was examined as a dichotomous variable, a cutoff value of the NLR for the risk of pouchitis was determined using ROC analysis. From the Youden index, the ROC analysis showed that the best cutoff value for the NLR was 2.15 (specificity: 67.7%, sensitivity: 72.2%) (Fig 2). Therefore, a cutoff value of 2.15 was chosen for further study. Twenty-six (53.1%) patients had an NLR of < 2.15 (low NLR group), whereas 23 (46.9%) patients had an NLR of ≥ 2.15 (high NLR group). Table 3 shows a comparison of the baseline characteristics between the low NLR (< 2.15) and high NLR (≥ 2.15) groups. No significant differences were noted in the background characteristics between the two groups. Fig 3 shows a comparison of the cumulative incidence of pouchitis between the low NLR group and the high NLR group. The incidence of pouchitis was significantly lower in the low NLR group than in the high NLR group (P = 0.01, log-rank test). Univariate Cox regression analysis also identified high NLR as a significant risk factor for the onset of pouchitis (unadjusted HR, 3.45; 95% CI, 1.23–9.71; P = 0.02). Therefore, to elucidate the influence of reported risk factors, Cox regression analysis

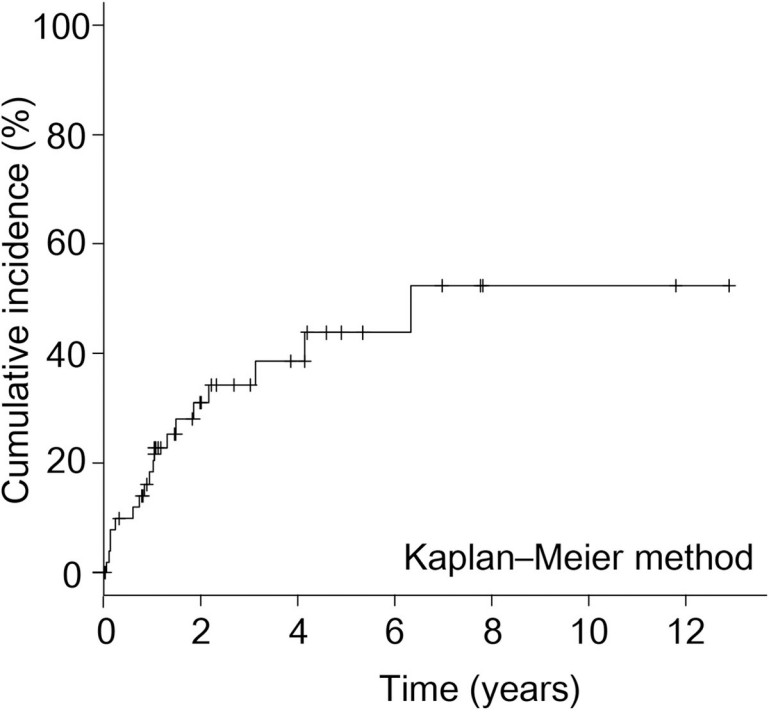

**Fig 1. Cumulative incidence of pouchitis.** The incidence of pouchitis was 19.2%, 32.6%, and 45.9% at 1, 2, and 5 years, respectively.

**Table 1. Baseline characteristics of the study population.**

|  | all patients |
| --- | --- |
| Number of patients | 49 |
| Sex: male/female | 25 / 24 |
| Age at diagnosis (years), median (interquartile range) | 32.4 (22.8–44.2) |
| Age at IPAA (years), median (interquartile range) | 44.3 (32.9–53.6) |
| Disease duration (years), median (interquartile range) | 4.5 (2.3–10.2) |
| UC location: Left-sided colitis/Pancolitis | 9 / 40 |
| Concomitant therapies, n (%) |  |
| Immunomodulator | 14 (28.6%) |
| Corticosteroids | 15 (30.6%) |
| Anti-TNF-α antibody therapy | 14 (28.6%) |
| Calcineurin inhibitor therapy | 16 (32.7%) |
| Hemoglobin (g/dL), median (interquartile range) | 12.9 (11.9–14.0) |
| Albumin (g/dL), median (interquartile range) | 4.20 (4.00–4.50) |
| CRP (mg/dL), median (interquartile range) | 0.07 (0.04–0.22) |
| WBC (/μL), median (interquartile range) | 5900 (5100–7200) |
| Neutrophil (/μL), median (interquartile range) | 3500 (2900–4600) |
| Lymphocyte (/μL), median (interquartile range) | 1700 (1400–2000) |
| NLR, median (interquartile range) | 2.00 (1.41–2.92) |
| Pouchitis (+), n (%) | 18 (36.7%) |

IPAA: ileal pouch-anal anastomosis; UC: ulcerative colitis; TNF: tumor necrosis factor; CRP: C-reactive protein; WBC: white blood cell; NLR: neutrophil-to-lymphocyte ratio.

**Table 2. Cox regression analysis of risk factors for the development of pouchitis.**

| | Unadjusted HR (95% CI) | P-value |
|---|---|---|
| Sex | | |
| Male | 1 | |
| Female | 0.76 (0.30–1.95) | 0.58 |
| Age at diagnosis (continuous) | 0.99 (0.95–1.02) | 0.47 |
| Age at IPAA (continuous) | 0.99 (0.95–1.02) | 0.40 |
| Disease duration (continuous) | 0.99 (0.94–1.05) | 0.81 |
| UC location | | |
| left-sided colitis | 1 | |
| pan-colitis | 3.98 (0.53–30.0) | 0.18 |
| Immunomodulators (azathioprine or 6-mercaptopurine) | | |
| No | 1 | |
| Yes | 1.29 (0.48–3.45) | 0.61 |
| Corticosteroids | | |
| No | 1 | |
| Yes | 0.65 (0.23–1.79) | 0.40 |
| Anti-TNF-α antibody therapy | | |
| No | 1 | |
| Yes | 2.21 (0.77–6.37) | 0.14 |
| Calcineurin inhibitor therapy | | |
| No | 1 | |
| Yes | 1.70 (0.66–4.43) | 0.27 |
| Albumin (continuous) | 0.38 (0.11–1.28) | 0.12 |
| CRP (continuous) | 1.23 (1.02–1.48) | 0.03 |
| NLR (continuous) | 1.14 (1.01–1.28) | 0.03 |
| Neutrophil (continuous, per 1000 /μL) | 1.24 (0.96–1.60) | 010 |
| Lymphocyte (continuous, per 1000 /μL) | 0.39 (0.17–0.89) | 0.02 |

HR: hazard ratio; CI: confidential interval; IPAA: ileal pouch-anal anastomosis; UC: ulcerative colitis; TNF: tumor necrosis factor; CRP: C-reactive protein; NLR: neutrophil-to-lymphocyte ratio.

was performed using the IPTW method to identify factors associated with the development of pouchitis. Variables included in the IPTW analysis were age, sex, disease location, and history of anti-TNF therapy. Cox regression analyses using the IPTW method identified NLR as a prognostic factor for the development of pouchitis by statistically adjusting for background factors (adjusted HR, 3.60; 95% CI, 1.31–9.89; P = 0.01).

As drug discontinuation during the course may occur in a number of situations, including a history of anti-TNF therapy as an adjusting background factor in IPTW analysis might not be appropriate in this study due to several biases. Therefore, we also performed IPTW analysis without a history of anti-TNF-therapy as an adjusting background factor. Cox regression analyses using the IPTW method, excluding the history of anti-TNF therapy as an adjusting background factor, also identified NLR as a prognostic factor for the development of pouchitis (adjusted HR, 3.67; 95% CI, 1.38–9.79; P = 0.01).

## Discussion

In this study, we investigated the utility of NLR for the development of pouchitis after IPAA in patients with UC. Our results suggest that a high NLR is strongly associated with an increased risk of developing pouchitis, and patients with high NLR should be followed up carefully.

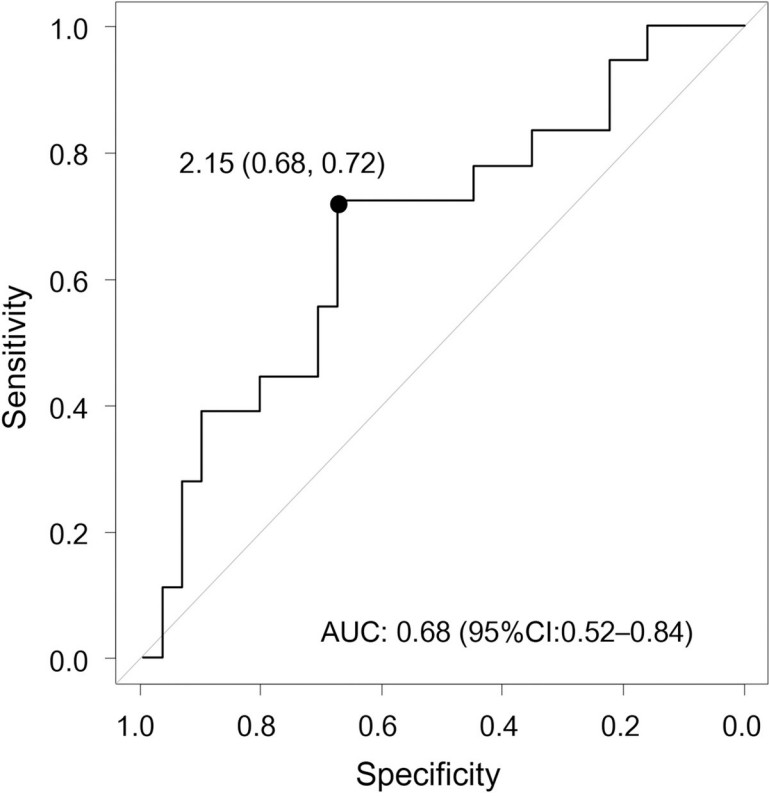

**Fig 2. Receiver operating characteristic curve for determining the cutoff value of the neutrophil-to-lymphocyte ratio (NLR) for predicting the development of pouchitis.** The optimal cutoff value for the NLR determined by maximal Youden's index was 2.15 (specificity: 67.7%, sensitivity: 72.2%). Area under curve (AUC): 0.68 (95% confidence interval [CI]: 0.52–0.84).

The mechanism by which the NLR predicts the development of pouchitis is unclear. Regarding neutrophils count, neutrophils are important leukocytes that can cause inflammation in UC [35]. Neutrophil accumulation and abscess formation within the intestinal crypts at the apical epithelial surface are typical pathological features of UC [36]. Patients with high peripheral neutrophil count may be prone to pouchitis. Indeed, patients who had higher neutrophil counts tended to develop pouchitis in this study, in line with a previous report by Koike et al. that identified a neutrophil percentage of > 65% before IPAA surgery as a risk factor for pouchitis [13]. Regarding lymphocyte count, Hirata et al. reported significantly increased numbers of CD19[+]CD138[+] cells in the pouchitis mucosa of patients with UC compared to non-inflamed UC pouches. The proliferation of this cell population suggests the possibility of involvement of a UC-derived abnormality in the pathogenesis of pouchitis [37]. The number of CD19[+]CD20[−]CD138[+] cells, which are an immature subset of IgG-producing plasma cells, was also significantly higher in the peripheral blood and inflamed colon of patients with active UC [38–40]. This cell subset was reported to have a feature of plasma cells; eccentrically located nuclei and abundant rough endoplasmic reticula, by immunoelectron microscopy [39]. Therefore, this cell population could be identified as a highly fluorescent population distinct from lymphocytes by a hematology analyzer, implying that UC patients with abundant CD19+CD138+ cells would have a relatively low percentage of lymphocytes in peripheral blood. This might be one of the possible mechanisms to explain how the NLR could predict the development of pouchitis.

**Table 3. Comparison between the low NLR group and high NLR group.**

|  | low NLR group | high NLR group | P-value |
|---|---|---|---|
| Number of patients | 26 | 23 |  |
| Sex: male/female | 12 / 14 | 13 / 10 | 0.57 |
| Age at diagnosis (years), median (interquartile range) | 36.4 (25.5–49.5) | 29.5 (21.5–43.8) | 0.12 |
| Age at IPAA (years), median (interquartile range) | 47.5 (38.6–55.6) | 35.9 (25.0–47.7) | 0.04 |
| Disease duration (years), median (interquartile range) | 4.2 (1.7–11.4) | 4.5 (3.1–9.5) | 0.87 |
| UC location: Left-sided colitis/Pancolitis | 4/22 | 5/18 | 0.72 |
| Concomitant therapies, n (%) |  |  |  |
| Immunomodulator | 6 (23.1%) | 8 (34.8%) | 0.53 |
| Corticosteroids | 7 (26.9%) | 8 (34.8%) | 0.76 |
| Anti-TNF-α antibody therapy | 6 (23.1%) | 9 (34.8%) | 0.53 |
| Calcineurin inhibitor therapy | 8 (30.8%) | 8 (34.8%) | 1 |
| Hemoglobin (g/dL), median (interquartile range) | 12.7 (11.8–13.5) | 12.9 (12.1–14.4) | 0.50 |
| Albumin (g/dL), median (interquartile range) | 4.10 (4.00–4.50) | 4.20 (3.90–4.40) | 0.86 |
| CRP (mg/dL), median (interquartile range) | 0.06 (0.04–0.24) | 0.07 (0.04–0.19) | 0.86 |
| WBC (/μL), median (interquartile range) | 5900 (4900–6800) | 5700 (5200–7700) | 0.44 |
| Neutrophil (/μL), median (interquartile range) | 5200 (4800–5600) | 4200 (3500–5000) | < 0.01 |
| Lymphocyte (/μL), median (interquartile range) | 2000 (1700–2400) | 1400 (1100–1700) | < 0.01 |
| NLR, median (interquartile range) | 1.47 (1.31–1.72) | 2.93 (2.46–4.15) | < 0.01 |
| Pouchitis (+), n (%) | 5 (19.2%) | 13 (56.5%) | < 0.01 |

IPAA: ileal pouch-anal anastomosis; UC: ulcerative colitis; TNF: tumor necrosis factor; CRP: C-reactive protein; WBC: white blood cell; NLR: neutrophil-to-lymphocyte ratio.

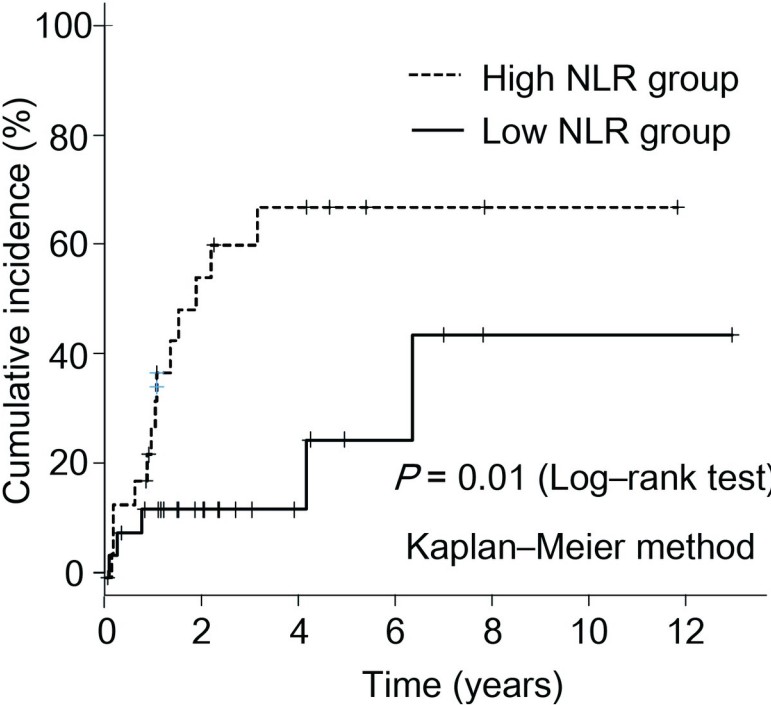

**Fig 3. Comparison of cumulative incidence of pouchitis between the high and low NLR groups.** The incidence of pouchitis was significantly lower in the low NLR group than in the high NLR group (P = 0.01, log-rank test).

Another possible mechanism is that peripheral WBCs might affect the gut microbiome. The role of the gut microbiome in pouchitis development is indicated by the effectiveness of antibiotics in the treatment of pouchitis, the use of probiotics for its prevention [41, 42], and the influence of innate lymphoid cells on the microbiome [43]. Taken together, patients with higher NLR might consist of a unique subset whose microbiome predisposes them to pouchitis.

Johnson et al. have reported that fecal calprotectin correlates with the severity of pouchitis and is a useful marker for diagnosing pouchitis [44]. Measuring fecal calprotectin is non-invasive, relatively cheap, and its sensitivity and specificity are very high. Although the fecal calprotectin test is a potentially useful screening tool for estimating the presence and severity of pouch inflammation, no evidence has been reported for its predictive role in inflammation. Regarding NLR, the NLR has been found to predict pouchitis before the operation.

Regarding medications before the operation, only a total steroid dose of $> 10000$ mg [13] and the use of infliximab [14] was reported to be a risk factor for pouchitis, to the best of our knowledge. In this study, although the use of immunomodulators, corticosteroids, anti-TNF-α antibody therapy, and calcineurin inhibitor therapy was not identified a risk factor for pouchitis, we could not say that these medications were not associated with the development of pouchitis because of the relatively small sample size and several biases such as drug discontinuation during the course.

As the sample size of this study was relatively small, we calculated the required sample size. Sample size calculation as a post-hoc analysis indicated that a total of 38 patients were required to detect a significant association between high NHR and the development of pouchitis with the following assumptions: an α level of 0.05, a β level of 0.20, half of the patients were allocated to the high NLR group, the incidences of pouchitis in patients with high and low NLR group were 54% and 19%, registration period was 13 years, and the follow-up period was 1.5 years. Therefore, the sample size in this study was satisfied to examine the association between NLR and the development of pouchitis.

This study has some limitations. First, this was a retrospective study with a relatively small cohort that is susceptible to bias in data selection and analysis. Second, we could not evaluate the predictive value of preoperative NLR since differential WBC count is not routinely measured in all patients just before the operation. Only 39 of 49 patients included in this study had their differential white blood cell count evaluated before the operation. Among these 40 patients, univariate Cox regression analysis did not identify NLR as a predictor of pouchitis development (HR, 0.98; 95% CI, 0.93–1.04; P = 0.56). A possible reason why preoperative NLR did not predict the development of pouchitis is that preoperative NLR would be affected by infection or treatment and, therefore, would not accurately reflect the situation. The other possible reason is that the number of patients analyzed for preoperative NLR was statistically too small. Therefore, we may not be able to conclude that the preoperative NLR is meaningless for pouchitis. Third, we were unable to evaluate the reported pouchitis predictive factors, such as extraintestinal manifestations of inflammatory bowel disease, preoperative terminal ileal inflammation, the presence of *interleukin-1 receptor antagonist gene allele 2*, total steroid dose of >10000 mg, or presence of perinuclear antineutrophil cytoplasmic antibodies owing to the retrospective design of the study.

Furthermore, NLR can be influenced by concurrent infections and concomitant drugs. Therefore, further large prospective studies will help confirm the NLR as a key predictor for pouchitis after IPAA in patients with UC.

Despite these limitations, our study suggests that the NLR could be associated with the development of pouchitis after IPAA in patients with UC. NLR should, therefore, be introduced in clinical practice.

## Supporting information

**S1 Dataset. Final analysis data of the study subjects.**
(CSV)

## Author Contributions

**Conceptualization:** Yu Nishida, Yasuhiro Fujiwara.

**Data curation:** Yu Nishida, Shuhei Hosomi, Hirokazu Yamagami, Satoko Ohfuji.

**Formal analysis:** Yu Nishida, Shuhei Hosomi, Satoko Ohfuji.

**Investigation:** Yu Nishida.

**Methodology:** Yu Nishida.

**Supervision:** Koji Fujimoto, Rieko Nakata, Shigehiro Itani, Yuji Nadatani, Shusei Fukunaga, Koji Otani, Fumio Tanaka, Yasuaki Nagami, Koichi Taira, Noriko Kamata, Toshio Watanabe, Yasuhito Iseki, Tatsunari Fukuoka, Masatsune Shibutani, Hisashi Nagahara, Yasuhiro Fujiwara.

**Visualization:** Yu Nishida.

**Writing – original draft:** Yu Nishida.

**Writing – review & editing:** Shuhei Hosomi, Yasuhiro Fujiwara.

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
