## [Decision Letter · Decision Letter 0]

8 Jul 2020

PONE-D-20-18412

Novel prognostic biomarkers of pouchitis after ileal pouch-anal anastomosis for ulcerative colitis: Neutrophil-to-lymphocyte ratio

PLOS ONE

Dear Dr. Hosomi,

Thank you for submitting your manuscript to PLOS ONE. After careful consideration, we feel that it has merit but does not fully meet PLOS ONE’s publication criteria as it currently stands. Therefore, we invite you to submit a revised version of the manuscript that addresses the points raised during the review process.

Your manuscript was assessed by two expert reviewers in this field, and some major issues have been raised by both reviewers as shown below.  Regarding the criticisms from both reviewers, you must fully address the issues in the revised results as well as discussion sections.  In particular, COX regression analysis for assessing incident risk over time seems inappropriate for this study, and therefore the authors need to provide the new data with more appropriate analysis and clear explanations in your revised manuscript.

We look forward to receiving your revised manuscript.

Kind regards,

Emiko Mizoguchi, M.D., Ph.D.

Academic Editor

PLOS ONE

Journal Requirements:

2. We noticed minor instances of text overlap with the following previous publication(s), which need to be addressed:

(1) https://journals.plos.org/plosone/article?id=10.1371%2Fjournal.pone.0213505

The text that needs to be addressed involves the Abstract (lines 21-24), Results section (lines 149-158) and the Discussion section (lines 175-191).

In your revision please ensure you cite all your sources (including your own works), and quote or rephrase any duplicated text outside the methods section. Further consideration is dependent on these concerns being addressed.

3. In your ethics statement in the manuscript and in the online submission form, please provide additional information about the patient records used in your retrospective study. Specifically, please ensure that you have discussed whether all data were fully anonymized before you accessed them.

Reviewers' comments:

Reviewer's Responses to Questions

**Comments to the Author**

1. Is the manuscript technically sound, and do the data support the conclusions?

Reviewer #1: Partly

Reviewer #2: Partly

2. Has the statistical analysis been performed appropriately and rigorously? 

Reviewer #1: No

Reviewer #2: No

3. Have the authors made all data underlying the findings in their manuscript fully available?

Reviewer #1: Yes

Reviewer #2: Yes

4. Is the manuscript presented in an intelligible fashion and written in standard English?

Reviewer #1: Yes

Reviewer #2: Yes

5. Review Comments to the Author

Reviewer #1: In this paper, the authors demonstrated that the incidence of pouchitis was significantly higher in the high NLR group than in the low NLR group. The authors concluded that NLR may be a useful biomarker for predicting the development of pouchitis after IPAA in UC. However, it is not unclear how NLR is involved in the development of pouchitis. I raise several concerns including this point as listed below.

1. Preoperative NLR was used in patients who have undergone one stage surgery, while postoperative NLR was used in those who have undergone two and three stage surgeries. To identify preoperative risk factors for the development of pouchitis, the authors should investigate the association between preoperative NRL and pouchitis.

2. Did the authors continue to use corticosteroids, immunmodulators, anti-TNF, and carcineurin inhibitor during the postoperative course? The authors used same method to analyze data in this study as previous paper (PLoS One. 2019; 14(3): e0213505.), however there are several biases as drug discontinuation during the course. Therefore, I think that cox regression analysis for assessing incident risk over time dose not fit in this study.

3. The authors showed that UC patients with high NLR have a high risk of pouchitis. However, I do not understand how NLR is involved in the development of pouchitis. Please discuss in more detail.

Reviewer #2: The authors proposed that neutrophil-to-lymphocyte ration is useful as an independent predicting marker of pouchitis after ileal pouch-anal anastomosis (IPAA) for ulcerative colitis.

However, they should address the following issues raised by this reviewer.

(1) On page 3, the introduction section, the authors had better mention not only the pouchitis-related disease and Neutrophil-to-lymphocyte ratio (NLR) but also diagnostic criteria of pouchitis and preexisting markers. This reviewer feels that the utility of the identified novel marker is difficult for readers to understand in the current manuscript.

(2) Twelve years ago, Johnson M.W. et al have been reported that fecal calprotectin correlates with the severity of pouchitis and is a useful marker (Eur J Gastroenterol Hepatol. 2008). Therefore, page 11, the discussion section, the authors should mention whether the identified NLR markers are useful markers of pouchitis compared to fecal calprotectin.

(3) Table.2, the authors used the multivariate regression with COX model analysis to identify the factors associated with the development of pouchitis. However, the number of cases may be insufficient to perform multivariate analysis using COX regression analysis in this study. Therefore, the authors should confirm the sample size required for COX regression analysis in this study.

(4) On page 5, line 4, the authors need to briefly mention the pouchitis disease activity index (PDAI).

(5) On page 8, second paragraph, line 5 to 8, the authors need to mention the statistical analysis method. 

(6) The authors should re-write the manuscript to make it easier for readers to understand. (e.g., page 8, first paragraph, the authors should mention the definition of high NLR and low NLR before mentioning that high NLR is a risk factor for pouchitis.)

6. PLOS authors have the option to publish the peer review history of their article (what does this mean?). If published, this will include your full peer review and any attached files.

Reviewer #1: No

Reviewer #2: No

---

## [Author Response · Author response to Decision Letter 0]

10 Aug 2020

August 11th, 2020

Dear Prof. Emiko Mizoguchi,

Academic Editor

PLOS ONE

We appreciate the opportunity to submit a revised version of our manuscript, titled “Novel prognostic biomarkers of pouchitis after ileal pouch-anal anastomosis for ulcerative colitis: Neutrophil-to-lymphocyte ratio” (manuscript ID; PONE-D-20-18412), for publication in PLOS ONE as an original article. We believe that we have addressed all comments raised by the reviewers, as detailed in the accompanying point-by-point reply.

We have also reanalyzed data and revised the manuscript and one table. 

We hope that these changes support and enhance our message: NLR may be a novel and useful indicator for predicting the development of pouchitis after ileal pouch-anal anastomosis in patients with ulcerative colitis.

The authors declare the following:

1. All authors concur with the submission of this revised manuscript.

2. This manuscript has not been published elsewhere in part or in its entirety and is not under consideration by another journal. This study design was approved by Osaka City University Hospital Certified Review Board, according to the Declaration of Helsinki. All the authors contributed to the manuscript and agree with this submission to your journal. 

3. I have read the journal's policy and the following authors of this manuscript have a competing interest: All authors declare that they have no conflicts of interest, and that there were no external funding sources for this study. This does not alter our adherence to PLOS ONE policies on sharing data and materials.

Many thanks for the opportunity to revise our manuscript. We hope that this manuscript is now acceptable for publication in PLOS ONE.

Sincerely yours,

Shuhei Hosomi M.D., Ph.D.

Department of Gastroenterology, Osaka City University Graduate School of Medicine

1-4-3, Asahi-machi, Abeno-ku, Osaka, 545-8585, Japan. 

Tel.: +81 6 6645 3811; fax: +81 6 6645 3813.

E-mail: m1265271@med.osaka-cu.ac.jp

Response to the journal requirements

Response to #1 requirement

We certify that our manuscript meets PLOS ONE’s style requirements.

Response to #2 requirement

We have rephrased these sentences: 

“Although there have been many investigations on neutrophil-to-lymphocyte ratio (NLR) in various diseases, its role in predicting the development of pouchitis remains unclear.” (lines 21–23)

“NLR was significantly associated with the development of pouchitis according to univariate Cox regression analysis (HR, 1.14; 95% CI, 1.01–1.28; P = 0.03). No other clinical variables such as age, sex, disease duration, age at onset, or disease location showed a statistically significant association with pouchitis development (Table 2).” (lines 154–157)

“In this study, we investigated the utility of NLR for development of pouchitis after IPAA in patients with UC. Our results suggest that a high NLR is strongly associated with an increased risk of developing pouchitis, and patients with high NLR should be followed up carefully.

The mechanism by which the NLR predicts the development of pouchitis is unclear. Regarding neutrophil counts, neutrophils are important leukocytes that can cause inflammation in UC [35].”(lines 194–198)

Response to #2 requirement

All data were fully anonymized before we accessed them. We have described this in the ethical consideration section:

“All data were fully anonymized before we accessed them.” (lines 130–131)

 

Response to the Reviewer #1 comments:

We appreciate your constructive comments and valuable suggestions. They have helped us to revise and improve this manuscript. As indicated in the responses that follow, we have considered these comments and suggestions in the revised version of our manuscript. All changes in the manuscript in response to the critiques are indicated with yellow highlights.

1. Preoperative NLR was used in patients who have undergone one stage surgery, while postoperative NLR was used in those who have undergone two and three stage surgeries. To identify preoperative risk factors for the development of pouchitis, the authors should investigate the association between preoperative NRL and pouchitis.

We are in full agreement that preoperative NLR should be investigated; however, in our hospital, differential white blood cell count was not measured in all patients before operation. We could not evaluate the predictive value of preoperative NLR sufficiently because of lack of data. Only 40 of 50 patients included in this study were evaluated for differential white blood cell count before operation. Among these 40 patients, univariate Cox regression analysis did not identify NLR as a predictor of development of pouchitis (hazard ratio, 0.98; 95% confidence interval, 0.93–1.04; P = 0.57). The possible reason why preoperative NLR did not predict development of pouchitis is that preoperative NLR would be affected by infection or treatment and would not reflect the situation. The other possible reason is that the number of patients in which preoperative NLR was analyzed was just too small. We cannot say preoperative NLR is meaningless for pouchitis; therefore, we have described this limitation in the Discussion: 

“we could not evaluate the predictive value of preoperative NLR since differential WBC count is not routinely measured in all patients just before operation. Only 40 of 50 patients included in this study had their differential white blood cell count evaluated before operation. Among these 40 patients, univariate Cox regression analysis did not identify NLR as a predictor of pouchitis development (HR, 0.98; 95% CI, 0.93–1.04; P = 0.57). A possible reason why preoperative NLR did not predict development of pouchitis is that preoperative NLR would be affected by infection or treatment and therefore would not accurately reflect the situation. The other possible reason is that the number of patients analyzed for preoperative NLR was statistically too small. Therefore, we may not be able to conclude that the preoperative NLR is meaningless for pouchitis.” (lines 225–234)

2. Did the authors continue to use corticosteroids, immunmodulators, anti-TNF, and calcineurin inhibitor during the postoperative course? The authors used same method to analyze data in this study as previous paper (PLoS One. 2019; 14(3): e0213505.), however there are several biases as drug discontinuation during the course. Therefore, I think that cox regression analysis for assessing incident risk over time dose not fit in this study.

We appreciate your valuable suggestions. As you suggested, multivariate Cox regression analysis of history of anti-TNF therapy for assessing incident risk over time may not fit in this study because of several biases. Reviewer 2 also suggested the number of cases may be insufficient to perform multivariate analysis using COX regression analysis in this study. Hence, we have performed Cox model analysis using inverse probability of treatment weighting (IPTW) [Sugihara M. et al. Pharmaceutical statistics. 2010;9(1):21-34] both including and excluding history of anti-TNF therapy for adjusting background factors, instead of performing multivariate Cox regression analysis. Analyses both including and excluding anti-TNF therapy history could identify NLR as a prognostic factor for the development of pouchitis. We have therefore described this in the Materials and Methods, as well as the Discussion: 

“The combined impact of the NLR and other known prognostic factors were investigated using Cox proportional hazard regression with inverse probability of treatment weighting (IPTW)” (lines 29–31)

“Cox regression analyses using IPTW also identified NLR as a prognostic factor for the development of pouchitis by statistically adjusting for background factors (HR, 3.43; 95% CI, 1.24–9.48; P = 0.02).” (lines 41–43)

“An IPTW analysis was applied to each observation in the Cox model to assess the relationship between the NLR and development of pouchitis. The IPTW analysis was derived using propensity scores on all observations before matching to reduce selection bias by statistically adjusting for background factors [34]. Variables included in the IPTW analysis were age, sex, disease location and history of anti-TNF therapy.” (lines 117–122)

“Therefore, to elucidate the influence of reported risk factors, Cox regression analysis was performed using the IPTW method to identify factors associated with the development of pouchitis. Variables included in the IPTW analysis were age, sex, disease location, and history of anti-TNF therapy. Cox regression analyses using the IPTW method identified NLR as a prognostic factor for the development of pouchitis by statistically adjusting for background factors (adjusted HR, 3.43; 95% CI, 1.24–9.48; P = 0.02). 

As drug discontinuation during the course may occur in a number of situations, including history of anti-TNF therapy as an adjusting background factor in IPTW analysis might not be appropriate in this study due to several biases. Therefore, we also performed IPTW analysis without history of anti TNF-therapy as an adjusting background factor. Cox regression analyses using the IPTW method, excluding history of anti-TNF therapy as an adjusting background factor, also identified NLR as a prognostic factor for the development of pouchitis (adjusted HR, 3.31; 95% CI, 1.23–8.94; P = 0.02).” (lines 169–180)

3. The authors showed that UC patients with high NLR have a high risk of pouchitis. However, I do not understand how NLR is involved in the development of pouchitis. Please discuss in more detail.

We agree that we should have included more detail on the possible mechanisms behind pouchitis development. As we described only the role of lymphocytes, we should have stated the role of neutrophils and the NLR. Neutrophils are important leukocytes that cause inflammation in UC [Hermanowicz A, et al. Clinical science 1985;69(3):241-9.]. Neutrophil accumulation and abscess formation within the intestinal crypts at the apical epithelial surface are typical pathological features of UC [Roche JK, et al. Clinical immunology and immunopathology. 1982;25(3):362-73.]. Patients with high peripheral neutrophil count may be prone to pouchitis. Regarding lymphocyte counts, Hirata et al. reported significantly increased numbers of CD19+Ki-67+ lymphocytes and CD138+Ki-67+ lymphocytes in the pouchitis mucosa of patients with UC compared to non-inflamed UC pouches. Proliferation of these lymphocytes suggests the possibility of involvement of a UC-derived abnormality in the pathogenesis of pouchitis [Hirata N, et al. Inflammatory bowel diseases. 2008;14(8):1084-90.]. The number of CD19+CD20–CD138+ lymphocytes in the peripheral blood was also significantly higher in patients with active UC [Hosomi S, et al. Clinical and experimental immunology. 2011;163(2):215-24.]; therefore, peripheral blood lymphocytes may be related to the development of pouchitis, which might explain how NLR could predict the development of pouchitis. 

Another possible mechanism is that peripheral WBCs might affect the microbiome in the gut. The role of the microbiome in the development of pouchitis is indicated by the effectiveness of antibiotics in the treatment of pouchitis, the use of probiotics for its prevention [Isaacs KL, et al. Inflammatory bowel diseases. 2007;13(10):1250-5.; Gionchetti P, et al. Gastroenterology. 2003;124(5):1202-9.], and the innate lymphoid cells influence on the microbiome [Thaiss CA, et al. Nature. 2016;535(7610):65-74.]. Taken together, patients with higher NLR might consist of a unique subset whose microbiome predisposes them to pouchitis. This is why patients with high neutrophil counts and with low lymphocyte counts are prone to pouchitis, and may explain how the NLR could predict the development of pouchitis. We described this mechanism in the Discussion section: 

“The mechanism by which the NLR predicts the development of pouchitis is unclear. Regarding neutrophil counts, neutrophils are important leukocytes that can cause inflammation in UC [35]. Neutrophil accumulation and abscess formation within the intestinal crypts at the apical epithelial surface are typical pathological features of UC [36]. Patients with high peripheral neutrophil count may be prone to pouchitis. Indeed, patients who had higher neutrophil counts tended to develop pouchitis in this study, in line with a previous report by Koike et al. that identified a neutrophil percentage of > 65% before IPAA surgery as a risk factor for pouchitis [13]. Regarding lymphocyte count, Hirata et al. reported significantly increased numbers of CD19+CD138+ cells in the pouchitis mucosa of patients with UC compared to non-inflamed UC pouches. Proliferation of this cell population suggests the possibility of involvement of a UC-derived abnormality in the pathogenesis of pouchitis [37]. The number of CD19+CD20–CD138+ cells, which are an immature subset of IgG-producing plasma cells, was also significantly higher in the peripheral blood and inflamed colon of patients with active UC [38-40]. This cell subset may be identified as a highly fluorescent population, distinct from lymphocytes, by an XE-5000 hematology analyzer. Therefore, a relatively low percentage of lymphocytes in peripheral blood may be related to the development of pouchitis, which might explain how the NLR could predict the development of pouchitis. 

Another possible mechanism is that peripheral WBCs might affect the gut microbiome. The role of the gut microbiome in pouchitis development is indicated by the effectiveness of antibiotics in the treatment of pouchitis, the use of probiotics for its prevention [41, 42], and the influence of innate lymphoid cells on the microbiome [43]. Taken together, patients with higher NLR might consist of a unique subset whose microbiome predisposes them to pouchitis.” (lines 197–217)

 

Response to the Reviewer #2 comments:

We appreciate your constructive comments and valuable suggestions that have helped improve this manuscript. As indicated in the responses that follow, we have considered these comments and suggestions, which are reflected in our revised manuscript. All changes in the manuscript made in response to the comments are highlighted in yellow.

(1) On page 3, the introduction section, the authors had better mention not only the pouchitis-related disease and Neutrophil-to-lymphocyte ratio (NLR) but also diagnostic criteria of pouchitis and preexisting markers. This reviewer feels that the utility of the identified novel marker is difficult for readers to understand in the current manuscript.

We appreciate your constructive comments. We should have stated preexisting markers as several studies have reported the utility of preexisting markers for predicting pouchitis, such as interleukin-1 receptor antagonist gene allele 2 [Carter MJ, et al. Gastroenterology. 2001;121(4):805-11.] and perinuclear antineutrophil cytoplasmic antibodies [Sandborn WJ, et al. The American journal of gastroenterology. 1995;90(5):740-7, Fleshner PR, et al. Gut. 2001;49(5):671-7.]. These two markers are useful for predicting pouchitis; however, measurement of these markers is not covered by medical insurance in Japan and is costly. Compared with these markers, the NLR is easily accessible and relatively cheap. We described this in the Introduction section. 

“Interleukin-1 receptor antagonist gene allele 2 and perinuclear antineutrophil cytoplasmic antibodies have been reported to be useful markers for predicting the development of pouchitis; however, measuring these markers is costly and not covered by medical insurance in Japan. Easily accessible and low-cost markers are required to predict pouchitis in patients with IPAA.” (lines 61 –65)

(2) Twelve years ago, Johnson M.W. et al have been reported that fecal calprotectin correlates with the severity of pouchitis and is a useful marker (Eur J Gastroenterol Hepatol. 2008). Therefore, page 11, the discussion section, the authors should mention whether the identified NLR markers are useful markers of pouchitis compared to fecal calprotectin. 

We appreciate your constructive comments. We should have stated the utility of calprotectin for assessment of pouch inflammation. Johnson et al. reported that fecal calprotectin correlates with pouchitis severity and is a useful marker. Measuring fecal calprotectin is non-invasive and relatively cheap, and its sensitivity and specificity for diagnosing pouchitis is very high. Although, the fecal calprotectin test is a potentially useful screening tool for estimating the existence and degree of pouch inflammation, no evidence has been reported for its predictive role in inflammation. Regarding NLR, the NLR could predict pouchitis before operation. Since the purposes of these tests are different, it would be necessary to use NLR and fecal calprotectin tests appropriately. We have explained this in the Discussion section: 

“Johnson et al. have reported that fecal calprotectin correlates with the severity of pouchitis and is a useful marker for diagnosing pouchitis [44]. Measuring fecal calprotectin is non-invasive, relatively cheap, and its sensitivity and specificity are very high. Although the fecal calprotectin test is a potentially useful screening tool for estimating the presence and severity of pouch inflammation, no evidence has been reported for its predictive role in inflammation. Regarding NLR, the NLR has been found to predict pouchitis before operation.” (lines 218–223)

(3) Table.2, the authors used the multivariate regression with COX model analysis to identify the factors associated with the development of pouchitis. However, the number of cases may be insufficient to perform multivariate analysis using COX regression analysis in this study. Therefore, the authors should confirm the sample size required for COX regression analysis in this study. 

We fully agree with your comment. We should not use the multivariate regression with COX model analysis to identify the factors associated with the development of pouchitis in this study due to the small sample size. Instead of performing multivariate Cox regression analysis, we attempted Cox model analysis using inverse probability of treatment weighting (IPTW) for unifying background factors. This analysis identified the NLR as a prognostic factor for the development of pouchitis (HR, 3.43; 95% CI, 1.24–9.48; P = 0.02). We have described this in the Materials and Methods and in the Discussion, and have revised Table 2 accordingly:

“An IPTW analysis was applied to each observation in the Cox model to assess the relationship between the NLR and development of pouchitis. The IPTW analysis was derived using propensity scores on all observations before matching to reduce selection bias by statistically adjusting for background factors [34]. Variables included in the IPTW analysis were age, sex, disease location and history of anti-TNF therapy.” (lines 118 –122)

“Therefore, to elucidate the influence of reported risk factors, Cox regression analysis was performed using the IPTW method to identify factors associated with the development of pouchitis. Variables included in the IPTW analysis were age, sex, disease location, and history of anti-TNF therapy. Cox regression analyses using the IPTW method identified NLR as a prognostic factor for the development of pouchitis by statistically adjusting for background factors (adjusted HR, 3.43; 95% CI, 1.24–9.48; P = 0.02). 

As drug discontinuation during the course may occur in a number of situations, including history of anti-TNF therapy as an adjusting background factor in IPTW analysis might not be appropriate in this study due to several biases. Therefore, we also performed IPTW analysis without history of anti TNF-therapy as an adjusting background factor. Cox regression analyses using the IPTW method, excluding history of anti-TNF therapy as an adjusting background factor, also identified NLR as a prognostic factor for the development of pouchitis (adjusted HR, 3.31; 95% CI, 1.23–8.94; P = 0.02).” (lines 169–180)

(4) On page 5, line 4, the authors need to briefly mention the pouchitis disease activity index (PDAI). 

We agree that we should have mentioned PDAI. We added a comment on PDAI in the Material and Methods section accordingly: 

“The pouchitis disease activity index (PDAI) is a 19-point index of pouchitis activity based on clinical symptoms as well as endoscopic and histologic findings [31]. In this study, pouchitis was diagnosed based on the modified pouchitis disease activity index (mPDAI) score using a combination of clinical symptoms and endoscopic examination. An mPDAI score of ≥ 5 points was used to define pouchitis in this study [32].” (lines 99–103)

 (5) On page 8, second paragraph, line 5 to 8, the authors need to mention the statistical analysis method. 

We thank the reviewer for the valuable suggestion. We revised the sentence in the Materials and Methods section and Result section: 

“Optimal cutoff values were determined according to the Youden criterion, which marks the point on a ROC curve where “sensitivity + specificity – 1” is maximal [33]. “(lines 112–114).

“From the Youden index, the ROC analysis showed that the best cutoff value for the NLR was 2.15 (specificity: 65.6%, sensitivity: 72.2%) (Fig 2).” (lines 159–160).

(6) The authors should re-write the manuscript to make it easier for readers to understand. (e.g., page 8, first paragraph, the authors should mention the definition of high NLR and low NLR before mentioning that high NLR is a risk factor for pouchitis.) 

We appreciate your valuable suggestion. We should re-write the manuscript to make it easier for readers to understand. We revised the Result section accordingly: “Table 3 shows a comparison of the baseline characteristics between the low NLR (< 2.15) and high NLR (≥ 2.15) groups. No significant differences were noted in the background characteristics between the two groups.” (lines 162–165)

---

## [Decision Letter · Decision Letter 1]

2 Sep 2020

PONE-D-20-18412R1

Novel prognostic biomarkers of pouchitis after ileal pouch-anal anastomosis for ulcerative colitis: Neutrophil-to-lymphocyte ratio

PLOS ONE

Dear Dr. Hosomi,

Thank you for submitting your manuscript to PLOS ONE. After careful consideration, we feel that it has merit but does not fully meet PLOS ONE’s publication criteria as it currently stands. Therefore, we invite you to submit a revised version of the manuscript that addresses the points raised during the review process.

Your manuscript was assessed by two reviewers same as the original manuscript, and some major issues still have been raised by both reviewers as shown below.  Regarding the criticisms, you must fully address the issues in the revised result and discussion sections.  In particular, as suggested by reviewer 1, the authors need to exclude the one-stage surgery data in the result section, if the authors decide to use post-operative data.  Without providing the proper data/explanations, this manuscript will not be considered for publication in PLOS ONE.

We look forward to receiving your revised manuscript.

Kind regards,

Emiko Mizoguchi, M.D., Ph.D.

Academic Editor

PLOS ONE

Reviewers' comments:

Reviewer's Responses to Questions

**Comments to the Author**

1. If the authors have adequately addressed your comments raised in a previous round of review and you feel that this manuscript is now acceptable for publication, you may indicate that here to bypass the “Comments to the Author” section, enter your conflict of interest statement in the “Confidential to Editor” section, and submit your "Accept" recommendation.

Reviewer #1: (No Response)

Reviewer #2: (No Response)

2. Is the manuscript technically sound, and do the data support the conclusions?

Reviewer #1: Partly

Reviewer #2: Yes

3. Has the statistical analysis been performed appropriately and rigorously? 

Reviewer #1: No

Reviewer #2: Yes

4. Have the authors made all data underlying the findings in their manuscript fully available?

Reviewer #1: Yes

Reviewer #2: Yes

5. Is the manuscript presented in an intelligible fashion and written in standard English?

Reviewer #1: Yes

Reviewer #2: Yes

6. Review Comments to the Author

Reviewer #1: The authors have not been able to adequately answer my questions. Please respond to the following questions.

1. I could not accept this result. The authors need to exclude the one-stage surgery data in the results, if the authors decide to use post-operative data. Or the authors need to increase the number of patients to show a significant difference using pre-operative data if the authors include the one-stage surgery data.

2. The authors need to prove that the number of patients is statistically sufficient in this study.

3. Why did the authors consider only TNF antibody as a background factor? There is a high possibility that steroids and immunomodulators are gradually being discontinued after surgery, so I think that it is necessary to consider them as background factors.

4. I agreed that the number of CD19+CD138+ cells, which are an immature subset of IgG-producing plasma cells, was significantly increased in the peripheral blood and inflamed colon of patients with active UC. However, I could not agree that a relatively low percentage of lymphocytes in peripheral. The authors need to show whether CD19+CD138+ cell subset is identified as a highly fluorescent population by an XE-5000 hematology analyzer. Or the authors try to explain why the lymphocytes are reduced in detail.

Reviewer #2: The authors have reanalyzed data and corrected the comments according to the reviewers, and this manuscript has been much improved. However, I strongly recommend to address the following issues raised by reviewer#1.

The reviewer#1 mentioned the continued use of corticosteroids, immunomodulators, calcineurin inhibitors, and anti-TNF therapy during the postoperative course. Because, there are several biases as drug discontinuation during the course. However, the authors have not addressed all of them in this manuscript. The authors would be better to analyze anti-TNF therapy as well as other therapies as background factors and briefly mentioned in the manuscript.

7. PLOS authors have the option to publish the peer review history of their article (what does this mean?). If published, this will include your full peer review and any attached files.

Reviewer #1: No

Reviewer #2: No

---

## [Author Response · Author response to Decision Letter 1]

5 Oct 2020

October 5th, 2020

Dear Prof. Emiko Mizoguchi,

Academic Editor

PLOS ONE

Dear Editor:

We appreciate the opportunity to submit a revised version of our manuscript, titled “Novel prognostic biomarkers of pouchitis after ileal pouch-anal anastomosis for ulcerative colitis: Neutrophil-to-lymphocyte ratio” (manuscript ID; PONE-D-20-18412), for publication in PLOS ONE as an original article. We believe that we have addressed all comments raised by the reviewers, as detailed in the accompanying point-by-point reply.

We have also reanalyzed the data and revised the manuscript and all the tables and figures with the cooperation of Satoko Ohfuji, who is a statistic expert in the Department of Public Health at our university. We have added her as a co-author. She declares that she has no conflicts of interest.

We hope that these changes support and enhance our message: NLR may be a novel and useful indicator for predicting the development of pouchitis after ileal pouch-anal anastomosis in patients with ulcerative colitis.

The authors declare the following:

1. All authors concur with the submission of this revised manuscript.

2. This manuscript has not been published elsewhere in part or in its entirety and is not under consideration by another journal. This study design was approved by Osaka City University Hospital Certified Review Board, according to the Declaration of Helsinki. All the authors contributed to the manuscript and agreed with this submission to your journal. 

3. I have read the journal’s policy and the following authors of this manuscript have a competing interest: All authors declare that they have no conflicts of interest and that there were no external funding sources for this study. This does not alter our adherence to PLOS ONE policies on sharing data and materials.

Many thanks for the opportunity to revise our manuscript. We hope that this manuscript is now acceptable for publication in PLOS ONE.

Sincerely yours,

Shuhei Hosomi, M.D., Ph.D.

Department of Gastroenterology, Osaka City University Graduate School of Medicine

1-4-3, Asahi-machi, Abeno-ku, Osaka, 545-8585, Japan. 

Tel.: +81 6 6645 3811; fax: +81 6 6645 3813.

E-mail: m1265271@med.osaka-cu.ac.jp

 

Response to the Reviewer #1 comments:

We appreciate your constructive comments and valuable suggestions as they helped us to revise and improve this manuscript. As indicated in the responses that follow, we have considered these comments and suggestions in the revised version of our manuscript. All changes in the manuscript in response to the critiques are indicated with yellow highlights.

1. I could not accept this result. The authors need to exclude the one-stage surgery data in the results if the authors decide to use postoperative data. Or the authors need to increase the number of patients to show a significant difference using pre-operative data if the authors include the one-stage surgery data.

We are in full agreement that we should have excluded the patients with one-stage surgery in the analysis. Among 50 patients included in this study, one-stage surgery was performed for only one patient. We excluded this patient and recalculated the data. The results did not change significantly. We revised all Figures and Tables and changed it to the recalculated value in the manuscript.

2. The authors need to prove that the number of patients is statistically sufficient in this study.

We are in full agreement that we should have mentioned whether the sample size is sufficient because the sample size in this study was rather small. We calculated the required sample size with the cooperation of Dr. Satoko Ohfuji, who is a statistic expert in the Department of Public Health at our university. Sample size calculation as a post-hoc analysis indicated that a total of 38 patients were required to detect a significant association between high NHR and the development of pouchitis with the following assumption: an α level of 0.05, a β level of 0.20, half of the patients were allocated to the high NLR group, the incidences of pouchitis in patients with high and low NLR group were 54% and 19%, registration period was 13 years, and the follow-up period was 1.5 years. Therefore, the sample size in this study was satisfied to examine the association between NLR and the development of pouchitis. We have added the following sentences in the Discussion. We have also added Dr. Satoko Ohfuji as a co-author. 

“As the sample size of this study was relatively small, we calculated the required sample size. Sample size calculation as a post-hoc analysis indicated that a total of 38 patients were required to detect a significant association between high NHR and the development of pouchitis with the following assumptions: an α level of 0.05, a β level of 0.20, half of the patients were allocated to the high NLR group, the incidences of pouchitis in patients with high and low NLR group were 54% and 19%, registration period was 13 years, and the follow-up period was 1.5 years. Therefore, the sample size in this study was satisfied to examine the association between NLR and the development of pouchitis.” (lines 233-239)

3. Why did the authors consider only TNF antibody as a background factor? There is a high possibility that steroids and immunomodulators are gradually being discontinued after surgery, so I think that it is necessary to consider them as background factors.

We appreciate your valuable suggestion. Several studies have reported the risk factors for pouchitis. Regarding medications before the operation, only a total steroid dose of > 10000 mg [Koike Y, et al. The Journal of surgical research. 2019; 238: 72-8.] and use of infliximab [Mor IJ, et al. Diseases of the colon and rectum. 2008; 51: 1202-7.] were reported to be risk factors for pouchitis, to the best of our knowledge. In this study, we focused only on anti-TNF-α antibody therapy, as we could not calculate the total steroid dose for some cases. We should have mentioned the influence of other medications on the development of pouchitis. Although the use of immunomodulators, corticosteroids, anti-TNF-α antibody therapy, and calcineurin inhibitor therapy was not identified as a risk factor for pouchitis, we could not say that these medications were not associated with the development of pouchitis. This is because there are several biases, such as drug discontinuation during the course and the sample size was relatively small. We rephrased the manuscripts as follows: “ Regarding medications before the operation, only a total steroid dose of > 10000 mg [13] and the use of infliximab [14] was reported to be a risk factor for pouchitis, to the best of our knowledge. In this study, although the use of immunomodulators, corticosteroids, anti-TNF-α antibody therapy, and calcineurin inhibitor therapy was not identified a risk factor for pouchitis, we could not say that these medications were not associated with the development of pouchitis because of the relatively small sample size and several biases such as drug discontinuation during the course.” (lines 227-232)

4. I agreed that the number of CD19+CD138+ cells, which are an immature subset of IgG-producing plasma cells, was significantly increased in the peripheral blood and inflamed colon of patients with active UC. However, I could not agree that a relatively low percentage of lymphocytes in peripheral. The authors need to show whether CD19+CD138+ cell subset is identified as a highly fluorescent population by an XE-5000 hematology analyzer. Or the authors try to explain why the lymphocytes are reduced in detail.

We are in full agreement that we should have mentioned the details of morphological features of the CD19+CD138+ cell subset. As shown in our department’s previous study, immunoelectron microscopy clearly showed that the CD19+CD138+ cells had a feature of plasma cells; eccentrically located nuclei and abundant rough endoplasmic reticula [Jinno Y. et al. Virchows Archiv 2006;448(4):412-21]. Therefore, this cell population could be identified as a highly fluorescent population distinct from lymphocytes by hematology analyzer, implying that UC patients with abundant CD19+CD138+ cells would have a relatively low percentage of lymphocytes in peripheral blood. We discussed this mechanism as one of the possible mechanisms to explain how NLR could predict the development of pouchitis. We rephrased the manuscript as follows: 

“This cell subset was reported to have a feature of plasma cells; eccentrically located nuclei and abundant rough endoplasmic reticula, by immunoelectron microscopy [39]. Therefore, this cell population could be identified as a highly fluorescent population distinct from lymphocytes by a hematology analyzer, implying that UC patients with abundant CD19+CD138+ cells would have a relatively low percentage of lymphocytes in peripheral blood. This might be one of the possible mechanisms to explain how the NLR could predict the development of pouchitis.” (lines 210-215)

Response to the Reviewer #2 comments:

We appreciate your constructive comments and valuable suggestions that have helped improve this manuscript. As indicated in the responses that follow, we have considered these comments and suggestions, which are reflected in our revised manuscript. All changes in the manuscript made in response to the comments are highlighted in yellow.

1. The reviewer#1 mentioned the continued use of corticosteroids, immunomodulators, calcineurin inhibitors, and anti-TNF therapy during the postoperative course. Because, there are several biases as drug discontinuation during the course. However, the authors have not addressed all of them in this manuscript. The authors would be better to analyze anti-TNF therapy as well as other therapies as background factors and briefly mentioned in the manuscript.

We are in full agreement that we should have mentioned other medications. As mentioned above, in this study, although the use of immunomodulators, corticosteroids, anti-TNF-α antibody therapy, and calcineurin inhibitor therapy was not identified as a risk factor for pouchitis, we could not say that these medications were not associated with the development of pouchitis because there are several biases such as drug discontinuation during the course and the sample size was relatively small. We rephrased the manuscript as follows: “Regarding medications before the operation, only a total steroid dose of > 10000 mg [13] and the use of infliximab [14] was reported to be a risk factor for pouchitis, to the best of our knowledge. In this study, although the use of immunomodulators, corticosteroids, anti-TNF-α antibody therapy, and calcineurin inhibitor therapy was not identified a risk factor for pouchitis, we could not say that these medications were not associated with the development of pouchitis because of the relatively small sample size and several biases such as drug discontinuation during the course.” (lines 227-232)

---

## [Editor Report · Decision Letter 2]

13 Oct 2020

Novel prognostic biomarkers of pouchitis after ileal pouch-anal anastomosis for ulcerative colitis: Neutrophil-to-lymphocyte ratio

PONE-D-20-18412R2

Dear Dr. Shuhei Hosomi;

We’re pleased to inform you that your manuscript has been judged scientifically suitable for publication and will be formally accepted for publication once it meets all outstanding technical requirements.

Kind regards,

Emiko Mizoguchi, M.D., Ph.D.

Academic Editor

PLOS ONE
---

## [Editor Report · Acceptance letter]

16 Oct 2020

PONE-D-20-18412R2 

Novel prognostic biomarkers of pouchitis after ileal pouch-anal anastomosis for ulcerative colitis: Neutrophil-to-lymphocyte ratio 

Dear Dr. Hosomi:

I'm pleased to inform you that your manuscript has been deemed suitable for publication in PLOS ONE. Congratulations! Your manuscript is now with our production department. 

Kind regards, 

on behalf of

Dr. Emiko Mizoguchi 

Academic Editor

PLOS ONE